# Evaluation of Signal Transducer and Activator of Transcription 3 (STAT-3) Protein Expression in Non-Hodgkin Lymphoma Cases in Hospital USM

**DOI:** 10.3390/diagnostics13091649

**Published:** 2023-05-07

**Authors:** Izyan Rifhana Muhamad, Noorul Balqis Che Ibrahim, Faezahtul Arbaeyah Hussain

**Affiliations:** 1Department of Pathology, School of Medical Sciences, Universiti Sains Malaysia Health Campus, Kubang Kerian 16150, Kelantan, Malaysia; 2Hospital Universiti Sains Malaysia, Universiti Sains Malaysia, Kubang Kerian 16150, Kelantan, Malaysia

**Keywords:** lymphoma, non-Hodgkin lymphoma, diffuse large B cell lymphoma, STAT-3 protein

## Abstract

Background: Evolving targeted therapy on Janus Associated Kinase-Signal Transducer and Activator of Transcription (JAK-STAT) signaling pathway, especially pertaining to STAT-3 protein in non-Hodgkin lymphoma (NHL), provides new treatment strategies. STAT-3 protein also relates to the prognostication of NHL. Hence, we aimed to evaluate the expression of STAT-3 protein in NHL cases diagnosed in Hospital Universiti Sains Malaysia (USM). Methods: A retrospective cross sectional study using formalin fixed paraffin embedded (FFPE) tissue blocks of 95 NHL cases were obtained. STAT-3 immunostaining was performed and evaluated. The proportion and association between the expression of STAT-3 protein with subtypes of NHL were statistically analyzed. Results: The majority of the cases (78.9%) had positive STAT-3 protein expression. 64.2% were among aggressive B cell NHL, whilst 20.0% of them were diffuse large B cell lymphoma, a non-germinal center B subtype (DLBCL-NGCB). There is also an association between STAT-3 protein expression with DLBCL subtypes (*p* = 0.046). Conclusion: Our study demonstrated a remarkable expression of STAT-3 protein in NHL, in which DLBCL subtypes had significant association. A larger scale study with a combination of JAK protein evaluation should be undertaken in the future.

## 1. Introduction

The pathogenesis of lymphoma is relatively complex and may involve a variety of gene mutations in the multistep origin of this cancer development. Mutations that increase tyrosine kinase activity are one of the common aberrations that drive cell growth [1]. Tyrosine kinase is an enzyme that mediates intracellular signaling by phosphorylating proteins on tyrosine residues to activate the Janus Associated Kinase-Signal Transducer and Activator of Transcription (JAK/STAT) signaling pathway [2]. The evolution of drugs targeting the JAK-STAT intracellular signalling pathway are becoming more recognized as drugs that can contribute to the treatment of haematolymphoid malignancies. Although JAK-STAT signaling is more well-known in hematological malignancies, more studies have been explored regarding its role in the pathogenesis of solid tumors, including lymphoma [3]. Four recognizable Janus-associated kinase (JAK) families are JAK-1, JAK-2, JAK-3, and TYK-2 proteins, while signal transducer and activator of transcription (STAT) families consist of seven major proteins, which are STAT-1, STAT-2, STAT-3, STAT-4, STAT-5a, STAT-5b, and STAT-6 [4].

The JAK-STAT pathway is an intracellular signaling pathway for gene expression mediated by cytokines in the form of interferon, interleukin, and growth factors [3]. The cytokine receptors receive extracellular signals once the cytokine molecules bind on them, resulting in their aggregation. The JAK protein, which is associated with cytokine receptors intracellularly, will phosphorylate and cause activation of the tyrosine kinase domain, providing docking sites for the STAT protein [3]. Recruited STAT monomers by the JAK-mediated phosphorylation result in their dimerization and translocation from the cell cytoplasm across the nuclear membrane to the cell nucleus. STAT dimers enhance the transcription of specific genes in the nucleus [3]. Changes in gene expression, cell proliferation and differentiation with the prevention of apoptosis occur because of this intracellular complex cascade of biochemical events of the JAK-STAT pathway [3,4,5], as shown in Figure 1. The development of some cancers as a result of genetic mutations, amplifications, or polymorphisms causes aberrant activation of the JAK-STAT pathway and leads to its continuous activation [3]. Other mechanisms that lead to the continuous activation of the JAK-STAT signaling pathways include extracellular signals by excessive cytokine expression, such as interleukin-6 upregulation by tumor cells and tumor microenvironment [6,7,8]. Therefore, the role of the aberrant activation of JAK-STAT signaling contributes to carcinogenesis, either as a tumor intrinsic driver of cancer growth/metastasis or as a modulator of immune surveillance [6].

In numerous human malignancies, phosphorylated tyrosine kinase and STAT nuclear expressions are signs of JAK-STAT pathway activation [3]. JAK-1 is known to be the main activator for STAT-3 activity, and excessive STAT-3 activation is common in lymphoma [7,9]. In fact, aberrant STAT-3 signaling plays an important role in immune evasion, and it becomes an oncogenic driver in several types of B-cell lymphoma and, also, in most T-cell lymphomas [7]. Targeting JAK/STAT3 signaling inhibits tumor growth and enhances anti-tumor immune responses [7]. In addition, previous studies agree that STAT-3 functions as an oncogene. Thus, inhibiting STAT-3 activity is an effective method for treating cancer. Aberrant STAT-3 activation has also been linked to cancer hallmarks and poor patient outcomes. Based on the outcomes of clinical trials, several STAT-3 inhibitors were well tolerated, and they may increase the overall survival of cancer patients, including lymphoma cases [8]. Therefore, aiming for JAK/STAT pathways in lymphoma has potential clinical value, and is a promising therapeutic target for treating the disease [7,10]. Thus, in line with the emerging therapeutic intervention targeting the JAK-STAT intracellular signaling pathway, focusing on STAT-3 expression in NHL types may provide some prognostic information. This study aims to evaluate the expression of STAT-3 protein, and to determine its prevalence and its associated clinicopathologic characteristics in NHL cases in Hospital Universiti Sains Malaysia (USM).

## 2. Materials and Methods

This retrospective cross-sectional study was carried out in the Department of Pathology, Hospital Universiti Sains Malaysia (Hospital USM). The study period was between January 2015 and December 2020. The keyword “lymphoma” was used to search the laboratory information system database. A total of 286 cases were reported within that period. However, only 95 of these cases were selected for this study, which were the ones that fulfilled the inclusion and exclusion criteria. The inclusion criteria included formalin fixed paraffin embedded (FFPE) tissue blocks that have adequate tumoral tissue, newly diagnosed cases, and complete clinical information from the patients’ records. Referral cases from other centers were excluded from this study. 

The 95 selected FFPE-tissue blocks were sectioned at 3 µm thickness. The STAT-3 primary antibody, recombinant rabbit monoclonal [EPR787Y] from ABCAM, was diluted at 1:200 dilution. Heat mediated antigen retrieval was performed using detection kit K8012 (DAKO) with high pH (pH9). Subsequently, all slides were placed in a manual stainer and incubated overnight at 4 °C. Endogenous peroxidase blocking, incubation with secondary antibody, and application of substrate chromogen and haematoxylin counterstaining in sequence were then applied. All samples were examined using an Olympus CX31 microscope at 400× magnification, with 10 high power field (HPF). STAT-3 staining can be seen in both the nuclear and the cytoplasm. In lymphoma cases, only those with nuclear staining were considered as positive (Figure 2) [11]. The scoring method (Table 1) was utilized [11].

Age, ethnicity, gender, the localization and size of tumors, B symptoms, lactate dehydrogenase (LDH) level, and the disease stage were included as clinicopathologic features in this study. Tumor histopathological characteristics were classified based on the WHO classification, revised 4th edition [12]. Data were analyzed using a Statistical Package for Social Sciences (SPSS) v. 27.0 software. Continuous data were expressed by mean and medium, while descriptive and categorical data were expressed by proportion, and Pearson chi-square and the Fisher-exact test with appropriate application. *p* < 0.05 were considered statistically significant.

## 3. Results

A total of 95 cases with NHL of various subtypes were selected, and the clinicopathologic characteristics are shown in Table 2 and Table 3. The mean age of this study was 54 years old. The ethnic was mainly among Malay (90.5%) ethnicity and males in gender (65.3%). Most of the patients had extra nodal site involvement (49.5%), a size of tumor less than 10 cm (63.2%), the presence of B symptoms (56.8%), a high lactate dehydrogenase (LDH) level (67.4%), and an advanced stage (III-IV) of disease (70.5%) during the first presentation. Many of them were diagnosed as aggressive B cell NHL (77.9%), largely contributed by DLBL subtypes (64.3% in total), and, specifically, DLBCL-NGCB subtypes (21.1%).

Amongst these 95 NHL cases, STAT-3 protein expressions were expressed in more than two-thirds (78.9%) of the cases, as shown in Table 4, using the scoring system with negative score in 20 cases, weak positive in 25 cases, and strong positive in 50 cases, respectively. Overall, STAT-3 protein expression was observed in 61 cases (64.2%) of aggressive B cell NHL, with 44 cases (88%) having a strong expression. The strong expressions were observed in DLBCL variants with 12 (24%) NOS, 15 (30%) NGCB, 6 (12%) GCB, 3 (6%) double expressions, and 1 (2%) triple expressions. The other aggressive type of B cell NHL is listed in Table 4, and had strong positive staining.

There was no significant association between STAT-3 protein expression with clinicopathologic characteristic age, ethnicity, and gender, nor the site and size of the tumors, B symptoms, LDH level, or disease staging (Table 5). Generally, there was no significant association, either, between STAT-3 protein expression with different NHL types based on cell lineage group and specific lymphoma subtypes. We categorized B cell NHL into an indolent subtype and an aggressive subtype. The later was further classified into two large groups: the DLBCL group and another aggressive B cell lymphoma group based on the WHO classification [12]. Among B cell NHL subtypes, we found that there was an association between DLBCL subtypes with STAT-3 protein expression (*p* = 0.046) (Table 6).

## 4. Discussion

This study involved 95 cases of NHL that have been diagnosed in a 5-year period in Hospital USM. The mean age of patients with NHL was 53 years old, which was almost similar to the median age (54 years old) among Asian populations, as reported in a previous study [13]. Younger age presentations across Asian populations were related to regional variation, endemic infection, and environmental exposure [14]. Lower prevalence of nodal type lymphoma was also hypothesized to be the contributing factor to the presentation of lymphoma in a younger population in Asian as compared to Western regions [13]. This was consistent with our study, whereby only a minority (31%) of patients had nodal site tumor involvement. Although the size of tumoral size was less than 10 cm (63.2%) in most of the cases, B symptoms (56.8%) experienced by the patients were postulated to be the reason why the patient came to seek the treatment in our center. Pyrexia of unknown origin (PUO) warranted the physician to do extensive clinical investigation, include a malignancy work up. 

NHL is so far consistently more common among males of Malay ethnicity, and patients tend to have an advanced stage of the disease during the initial presentations, which were also observed in our study population, as reported by Ab Manan et al. [15]. Since the advanced stage of the disease was frequently observed among our patients, the baseline high LDH levels of ≥480 U/L (67.4%) reflecting the high tumor burden was expected.

In our study, the majority (78.9%) of NHL cases expressed STAT-3 protein. Comparing different NHL subtypes, aggressive B cell lymphomas (64.2%) were predominantly the STAT-3 protein expression, especially DLBCL subtypes (54.7%). Most NHL subtypes had no correlation with STAT-3 expression. However, when we specifically focused on B cell NHL subtypes, there was significant association between STAT-3 protein expression with DLBCL subtypes (*p* = 0.046).

Between B cell NHL, Ohgami et al. found the STAT-3 expression was highest among DLBL (36%) compared to other subtypes [16]. Furthermore, in several studies focusing only on DLBCL, 41% to 84% of cases showed STAT-3 expression in which high STAT-3 expression was observed in 25.7% to 46% of the cases [11,17,18,19]. While Wu et al. and Kwon et al. identified 32% and 83% of the DLBCL-NGCB subtype expressed STAT-3 protein, respectively, our study also demonstrated that the DLBCL-NGCB subtype was the highest (20%) among DLBCL subtypes [17,19]. Among DLBCL subtypes, a higher percentage of DLBCL-NGCB subtype (83%) was associated with STAT-3 expression [19]. Concerning STAT-3 expression in DLBCL, it was related to concurrent downregulation of BCL-6 and MYC, which leads to STAT-3 activation, especially those with the DLBCL-NGCB subtype [16,19]. The current study agrees with previous studies, as epidemiological data of DLBCL subtypes remain the major NHL subtypes throughout the world [13]. Therefore, there has been much interest in DLBCL subtypes, and they have been explored.

Apart from DLBCL, there was little difference between other cases of lymphoma subtypes that did not express STAT-3 and those that did. Interestingly, STAT-3 expressions were present in all three cases of mantle cell lymphomas (MCL) and three out of four primary mediastinal large B cell lymphoma (PMBL) cases. These were contradicted with the findings of previous studies of STAT-3 expression in PMBL or MCL [16]. Our study also found that most T/Natural Killer cells NHL expressed STAT-3, apart from Anaplastic Large cell lymphoma (ALCL) and Anaplastic Lymphoma Kinase-positive (ALK-positive). The latter barely accounted for one in three ALCL cases. However, there is still a lack of work on using IHC technique for comparison. Nevertheless, earlier molecular studies do show that none to a very small proportion of ALCL ALK-positive cases have *STAT3* gene mutation [20,21]. In contrast, a wide range (6% to 72.4%) of Extra nodal Natural Killer/T cell lymphoma nasal type cases harbored the *STAT3* gene mutation [7,20,22,23,24]. In fact, STAT-3 IHC was reliable as a surrogate marker in routine samples [11,17]. There was no previous study pertaining to the expression of STAT-3 protein in T-lymphoblastic leukemia/lymphoma for comparison.

No correlation has been found between STAT-3 expression and the clinicopathological parameters included in this study, which agreed with the studies completed by Wu et al. and also by Kwon et al. [17,19]. However, in the most recent studies, STAT-3 activity was also associated with advanced stage (stage III/IV), multiple extra-nodal sites involvement, the advanced age of the patient, the presence of B symptoms (presence of fever, weight loss, night sweat), bone marrow involvement, bulky lymph nodes, and a poor prognosis in DLBCL [11,18,25]. These studies were comparable to our study in terms of the sample size, the IHC technique, and the interpretation of the result. The difference was that our study compared the association of both non-expressed STAT-3 and expressed STAT-3 cases rather than comparing weak- and strong STAT-3 expressions. Therefore, there was discrepancy between the two groups of researchers.

There are several limitations to our study. Only a small sample size (95) of NHL cases were qualified in this study in a single institutional center. Furthermore, the initial purpose of this study was to investigate in detail the overall JAK-STAT signaling pathway that will contribute the general understanding of lymphomagenesis. However, due to time constraints, the limitation of the budget, and the difficulty of optimizing the JAK-2 primary antibody, only a part of the initial objective was successfully fulfilled. In future, we hope that a study akin to this can be undertaken on a larger scale, with a combination of JAK protein evaluation by collaborating with other institutions and companies in order to establish more significant findings.

## 5. Conclusions

Our study demonstrated more frequent expression of STAT-3 protein in NHL, in which DLBCL subtypes are statistically significant (*p* = 0.046). These observations may give additional local data to aid in the understanding of the involvement of this protein in NHLs.

## Figures and Tables

**Figure 1 diagnostics-13-01649-f001:**
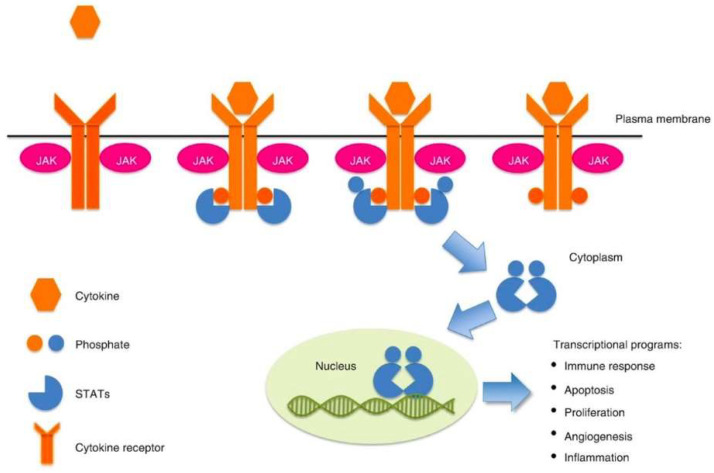
JAK-STAT signaling pathway [3].

**Figure 2 diagnostics-13-01649-f002:**
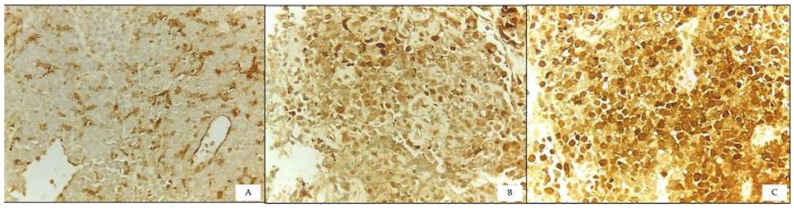
Expression of STAT-3 immunostaining: (**A**) Weak nuclear staining, (**B**) Moderate nuclear staining, and (**C**) Strong nuclear staining (400× magnification).

**Table 1 diagnostics-13-01649-t001:** Scoring system for STAT3 stain [11].

Score	Extent of Markers Expression	Intensity of Positive Score
0	No positive cells/HPF	Nil
1	<10% positive cells/HPF	Weak
2	10–30% positive cells/HPF	Moderate
3	>30% positive cells/HPF	Strong

Score = Intensity × Extent. (Values: 0 to 9, >3 is high expression; <3 is low expression; 0 is negative). HPF: high power field.

**Table 2 diagnostics-13-01649-t002:** Demographic and clinicopathological characteristics of patients (*n* = 95).

			*n*	(%)
Age Group (years)	≤60	54 (mean)	51	(53.7)
	>60		44	(46.3)
Ethnic group	Malay		86	(90.5)
	Non-Malay			
	- Chinese		7	(7.4)
	- Indian		1	(1.1)
	- Others		1	(1.1)
Gender	Female		33	(34.7)
	Male		62	(65.3)
Site	Nodal		30	(31.6)
	Extra nodal		47	(49.5)
	Nodal and extra nodal		18	(18.9)
Tumour size (cm)	<10	7 (median)	60	(63.2)
	≥10		35	(36.8)
B symptoms	Absent		41	(43.2)
	Present		54	(56.8)
LDH level (U/L)	<480 (Normal)	720 (median)	31	(32.6)
	≥480 (Elevated)		64	(67.4)
Stage of disease	Stage I–II (Early)		28	(29.5)
	Stage III–IV (Advanced)		67	(70.5)

**Table 3 diagnostics-13-01649-t003:** NHL histopathological subtypes (*n* = 95).

		*n*	(%)
Type	T/Natural Killer cells	7	(7.4)
	B cells	88	(92.6)
Subtypes	Indolent B cell	14	(14.7)
	Follicular lymphoma	4	(4.2)
	Mantle cell lymphoma	3	(3.2)
	Extra nodal Marginal Zone Lymphoma	3	(3.2)
	Lymphoplasmacytic lymphoma	1	(1.1)
	Primary cutaneous follicular centre lymphoma	1	(1.1)
	SLL	2	(2.1)
	Aggressive B cell DLBCL group	74 61	(77.9) (64.3)
	- DLBCL-NGCB	20	(21.1)
	- DLBCL-GCB	15	(15.8)
	- DLBCL NOS	17	(17.9)
	- DLBCL double expressions	7	(7.4)
	- DLBCL triple expressions	2	(2.1)
	Others aggressive B cell	13	(14.0)
	- PMBL	4	(4.2)
	- Primary DLBCL of CNS	3	(3.2)
	- EBV-positive DLBCL, NOS	1	(1.1)
	- T cell/histiocyte- rich B cell lymphoma	1	(1.1)
	- B-cell lymphoma, unclassifiable, with features intermediate between diffuse large B-cell lymphoma and classic Hodgkin lymphoma	1	(1.1)
	- Plasmablastic lymphoma	1	(1.1)
	- B-lymphoblastic leukaemia/lymphoma	1	(1.1)
	- Burkitt Lymphoma	1	(1.1)
	T cell NHL	7	(7.4)
	T-lymphoblastic leukaemia/lymphoma	3	(3.2)
	ALCL, ALK-positive	3	(3.2)
	Extra nodal Natural Killer/T cell lymphoma, nasal type	1	(1.1)

Abbreviation: DLBCL: Diffuse large B cell lymphoma; DLBCL-NGCB: Diffuse large B cell lymphoma, Non-Germinal Centre B subtypes; DLBCL-GCB: Diffuse large B cell lymphoma, Germinal Centre B subtypes; NOS: Not Otherwise Specified; CNS: Central Nervous System; PMBL: Primary Mediastinal Large B Cell Lymphoma.

**Table 4 diagnostics-13-01649-t004:** The expression of STAT-3 protein in NHL histopathological subtypes (*n* = 95).

Subtypes		STAT-3 Protein Expression	
		Negative	Positive
		*n* (%)	*n* (%)
B cell NHL	Indolent B cell	5 (5.3)	9 (9.5)
	Follicular lymphoma	2 (2.1)	2 (2.1)
	Mantle cell lymphoma	0 (0.0)	3 (3.2)
	Extra nodal Marginal Zone Lymphoma	1 (1.1)	2 (2.1)
	Lymphoplasmacytic lymphoma	1 (1.1)	0 (0.0)
	Primary cutaneous follicular centre lymphoma	0 (0.0)	1 (1.1)
	SLL	1 (1.1)	1 (1.1)
	Aggressive B cell	13 (13.7)	61 (64.2)
	1. DLBCL group	9 (9.5)	52 (54.7)
	- DLBCL-NGCB	1 (1.1)	19 (20.0)
	- DLBCL-GCB	4 (4.2)	11 (11.6)
	- DLBCL NOS	2 (2.1)	15 (15.8)
	- DLBCL double expressions	2 (2.1)	5 (5.3)
	- DLBCL triple expressions	0 (0.0)	2 (2.1)
	2. Others aggressive B cell	4 (4.2)	9 (9.5)
	- PMBL	1 (1.1)	3 (3.2)
	- Primary DLBCL of CNS	1 (1.1)	2 (2.1)
	- EBV-positive DLBCL, NOS	0 (0.0)	1 (1.1)
	- T cell/histiocyte- rich B cell lymphoma	0 (0.0)	1 (1.1)
	- B-cell lymphoma, unclassifiable, with features intermediate between diffuse large B-cell lymphoma and classic Hodgkin lymphoma	0 (0.0)	1 (1.1)
	- Plasmablastic lymphoma	1 (1.1)	0 (0.0)
	- B-lymphoblastic leukaemia/lymphoma	1 (1.1)	0 (0.0)
	- Burkitt Lymphoma	0 (0.0)	1 (1.1)
T cell NHL	T cell	2 (2.1)	5 (5.3)
	T-lymphoblastic leukaemia/lymphoma	0 (0.0)	3 (3.2)
	ALCL, ALK-positive	2 (2.1)	1 (1.1)
	Extra nodal Natural Killer/T cell lymphoma, nasal type	0 (0.0)	1 (1.1)
Total	95 (100.0)	20 (21.1)	75 (78.9)

Abbreviation: DLBCL: Diffuse large B cell lymphoma; DLBCL-NGCB: Diffuse large B cell lymphoma, Non-Germinal Centre B subtypes; DLBCL-GCB: Diffuse large B cell lymphoma, Germinal Centre B subtypes; NOS: Not Otherwise Specified; CNS: Central Nervous System; PMBL: Primary Mediastinal Large B Cell Lymphoma.

**Table 5 diagnostics-13-01649-t005:** The association between clinicopathologic characteristics with STAT-3 protein expression in NHL (*n* = 95).

		STAT-3 Protein Expression				
		Negative		Positive		*p* Value
		*n*	(%)	*n*	(%)	
Age group (years)	≤60	11	(55.0)	40	(53.3)	0.894 ^a^
	>60	9	(45.0)	35	(46.7)	
Ethnic group	Malay	18	(90.0)	68	(90.7)	>0.950 ^b^
	Non-Malay	2	(10.0)	7	(9.3)	
Gender	Female	7	(35.0)	26	(34.7)	0.978 ^a^
	Male	13	(65.0)	49	(65.3)	
Site	Nodal	3	(15.0)	27	(36.0)	0.196 ^a^
	Extra nodal	12	(60.0)	35	(46.7)	
	Nodal and extra nodal	5	(25.0)	13	(17.3)	
Tumour size (cm)	<10	10	(50.0)	50	(66.7)	0.170 ^a^
	≥10	10	(50.0)	25	(33.3)	
B symptoms	Absent	11	(55.0)	30	(40.0)	0.229 ^a^
	Present	9	(45.0)	45	(60.0)	
LDH level (U/L)	<480 (Normal)	6	(30.0)	25	(33.3)	0.778 ^a^
	≥480 (Elevated)	14	(70.0)	50	(66.7)	
Stage of disease	Stage I-II (Early)	7	(35.0)	21	(28.0)	0.542 ^a^
	Stage III-IV (Advanced)	13	(65.0)	54	(72.0)	

^a^ Pearson Chi Square test. ^b^ Fisher’s exact test.

**Table 6 diagnostics-13-01649-t006:** The association between NHL type/subtypes with STAT-3 protein expression (*n* = 95).

		STAT-3 Protein Expression				
		Negative		Positive		*p* Value
		*n*	(%)	*n*	(%)	
Type	T/Natural Killer cells	2	(10.0)	5	(6.7)	0.636 ^b^
	B	18	(90.0)	70	(93.3)	
Subtype	Indolent B cell	5	(25.0)	9	(12.0)	0.214 ^b^
	Aggressive B cell	13	(65.0)	61	(81.3)	
	T/Natural Killer cells	2	(10.0)	5	(6.7)	
B cell NHL	DLBCL group	9	(50)	52	(74.3)	0.046 ^a^
	Others B cell NHL	9	(50)	18	(25.7)	

^a^ Pearson Chi Square test. ^b^ Fisher’s exact test.

## Data Availability

Not applicable.

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
