# Peer review of "Evaluation of Signal Transducer and Activator of Transcription 3 (STAT-3) Protein Expression in Non-Hodgkin Lymphoma Cases in Hospital USM"

_diagnostics, 2023, doi:10.3390/diagnostics13091649_

Round 1

Reviewer 1 Report

Thank you very much for providing an opportunity to review the article titled “Evaluation of Signal Transducer and Activator of Transcription 3 (STAT-3) Protein Expression in non-Hodgkin Lymphoma Cases in Hospital USM” by Izyan Rifhana Muhamad and co-authors.

In this manuscript, Izyan and co-authors have performed a retrospective investigation about the expression of STAT3 protein using the FFPE tissue blocks from 95 NHL cases diagnosed in Hospital Universiti Sains Malaysia. The authors shows that 78% of the analyzed samples expressed STAT3, among which 64% were aggressive lymphomas whilst 20% of them belongs to DLBCL-NGCB subtype. The authors report a statistical significant association of STAT3 expression with DLBCL group. 

Major comments:

  1. The authors classified the extend of STAT3 expression as weak, moderate and strong (scoring system). What are the number of cases in each intensity groups?
  2. How does the extend of the STAT3 expression associates with NHL subtypes? Is there any positive correlation with the extend of STAT3 expression with disease aggressiveness or different subtypes of DLBCL? (For example does the strong expressors belongs to aggressive subtypes like NGCB, double expressor or triple expressor).

Reviewer 2 Report

The authors present a retrospective study investigating the expression of STAT-3 protein in lymphoma samples. Considering the growth of therapies aiming the JAK-STAT pathways over the last decade, it is of clinical importance to understand how this pathway is implicated in other diseases and if it correlates with clinical characteristics. Considering this, the study is clinically relevant and deserves attention. Nevertheless, the manuscript needs extensive English language review. In addition, there is only a small amount amount of clinical information, which decreases the relevance of this manuscript. Gathering and presenting more clinical information could be key for better understanding of this pathway in lymphoma.

Please check the comments below for further improvement suggestions.

Please consider an extensive language review. There are several conjugation and syntax errors, which in some cases difficult the understanding of the manuscript.

Please consider adding a figure explaining the JAK-STAT signaling pathway. 

Line 69: "Many researchers agree" is not a strong argument. Please rephrase endorsing the rationale that supports this hypothesis.

Did you have exclusion criteria? If yes please mention them in the Methods section 

Did you collect data regarding previous therapy and/or if the samples are from patients with recidiv/relapse? 

The first two paragraphs of the discussion are not important to the general understanding of the manuscript. I would suggest using these to summarize the most important findings of the manuscript concerning STAT-3, which is the main focus.

Does STAT-3 correlate with outcomes? What about overall mortality, PFS? Does it correlate with responsiveness to chemotherapy? These would be particularly important in DLBCL in which the prevalence of STAT-3 expreassion was higher. 

In the Methods section the authors indicate a scoring system concerning the strength of expression, but it is no further mentio of it throughout the study. I believe it would be very interesting to investigate the strength of expression in the different lymphoma subtypes and if there is relevant differences in multiple DLBCL patients

Reviewer 3 Report

Line 28: "complex" not complexed.  Line 31: "phosphorylating" not phosphorylates.  Lines 33-37: The sentence should be clarified, it is confusing as it is written. Line 43: add "pathway" after signaling.  Line 46:  add "is" after which.  Line 47 : should be "phosphorylate"  "cause". Line 50: should be "enhance". Lines 5: "cause" not causing. Line 57: "include" not includes. Line 64:  "after to be' add "the". Line 66: "becomes" not become. Line 74:  "pathways" instead of pathway (unless it is singular, then "has" potential .  Line 79: "expression" not expressions. Line 80:  Add Comma, after prevalence. Line : 86  "a total of 286 cases were reported within that period". Line 87: selected "for" rather than in. Line 94:  placed in "a" manual. Line 120 : "patients" instead of patient. Line 161 : in "a" younger population in Asia "as compared to Western regions".  Line 162: only "a" minority; of "patients" "had " rather than came with. Line 174 : "predominantly". Line 182 : references for "Wu et al", and "Kwon et al".  Line 220 : "purpose" not propose. Line 224 : extended "to" better than "in".

This manuscript was a rather difficult one to review. Whereas, the subject is valid and important, the presentation is difficult to follow at times; some sentences are too long. This takes away some of the importance of the manuscript, and one has to go back and forth to understand the intended meaning. Also, there is a great deal of research on this topic, and I would like to see additional recent references. There is only one reference from 2022, some from 2019, but others were older. 

Though I did not see any problems with the Figure or the Tables.

Although this manuscript addresses patients mainly in Malaysia. I would like to see some changes and possibly additional results to enhance the significance of this manuscript. I will be more convinced to accept this. manuscript.

Round 2

Reviewer 3 Report

The authors significantly improved the manuscript.

They added a Figure (Figure 1), depicting the JAK-STAT signaling pathway. However, a significant problem "There is now Figures 1 and 3, but no Figure 2. I am presuming that they mislabeled Figure 3, which now should be Figure 2.  If That is the case, then Figure 3 should be changed to Figure 2, and the text referring to Figure 3 (line 105), should be changed to Figure 2. Check other text that may refer to Figure 3, and change it to Figure 2. Otherwise, insert Figure 2. Also line 105 usually reference to Tables usually should be Capitalized (Table 2). If there is an Figure 2, then that should be included.

Line 139 "two-thirds" 

I recommend adding the number of a reference even though it is in text. It is easier for a reader to look-up the reference by the number assigned, rather than go through the list of references to find it, which may take time, particularly if the an author is referenced more than once. Examples are references on lIne 1921, and line 194; check for others.

I believe that with these changes, This manuscript would be ready to be published.
